# Investigation on the Coalescence of Filaments in Fused Filament Fabrication Process for Amorphous Polyether Ketone Ketone for Varying Nozzle Diameters and Chamber Temperatures

**DOI:** 10.3390/polym16192732

**Published:** 2024-09-27

**Authors:** Antoine Runacher, Thomas Joffre, Germain Fauny, Claudia Salvan, Nils Marchal, Francisco Chinesta

**Affiliations:** 1Centre Technique Innovation Plasturgie et Composites, 2 Rue Pierre et Marie Curie, 01100 Bellignat, France; nils.marchal@ct-ipc.com (N.M.); germain.fauny@ct-ipc.com (G.F.); claudia.salvan@ct-ipc.com (C.S.); thomas.joffre@ct-ipc.com (T.J.); francisco.chinesta@ensam.eu (F.C.); 2Laboratoire PIMM, Ecole Nationale des Arts et Métiers, 151 Boulevard de l’Hôpital, 75013 Paris, France

**Keywords:** fused filament fabrication, high-performance polymer, filament modelling, sintering, coalescence, microscope imaging, nozzle diameter, cooling temperature, mesostructure, porosity

## Abstract

The use of Fused Filament Fabrication (FFF) of high-performance polymers is becoming increasingly prevalent, leading to the exploration of new applications. The use of such materials in critical cases for aerospace applications necessitates the verification of industry standards, particularly with regard to the requirements for part porosity. The authors investigate the effect of nozzle diameter and cooling temperature printing parameters on the porosity of the part by using existing modelling methods based on the sintering of cylinders and spheres and comparing the results to microscope snapshots of sections of parts. The models are able to be used as limits for predicting the longitudinal neck growth of the part. The authors show through experiments that the value of the cooling temperature of the deposited filament has a minimal effect on the outcome, while nozzle diameter has a strong impact on the resulting porosity. The modelling results show that there is a significant impact of both the nozzle diameter and cooling temperature on the porosity of the part. This implies that further refinement of the models is needed for the resulting parts to be applied in critical structures.

## 1. Introduction

Since its birth in the 1960s, Additive Manufacturing (AM) methods have been rapidly developing. While not as efficient as traditional subtractive and formative methods, the quick design-to-part time and the customisation possibilities of AM have contributed to its development, first for prototyping endeavours and now even for structural uses of parts [1].

A large number of such methods were developed over the years for metals and polymer materials, each having its own drawbacks and perks. This multitude of methods has resulted in a rapid increase in scientific publications regarding this set of methods and also contributed to the expansion of the field [2].

One of these fabrication processes is Fused Filament Fabrication (FFF) [3]. By feeding a polymer through a heating nozzle, a melted filament of material is extruded following a specific path [4]. Adjacent filaments will fuse to form a solid ensemble after the cooling of the material [5]. Successive layers of material deposited on top of each other will then form a finished part. This process can be applied to a wide array of materials [6,7,8] to produce parts of complex geometry, without moulds or welds. It is, however, slower than traditional manufacturing, the manufactured parts are limited in size, and their mechanical properties are often worse than parts fabricated with the injection process [9].

There exists a market for FFF-manufactured parts: while the manufacturing times are high, there are no added costs of moulds. For a low volume of parts, these characteristics can be more advantageous than injection or welding. This is particularly the case in the aviation industry, as the parts made using FFF are lighter than their aluminium counterparts used in planes today and boast very low excess material consumption. Still, the high safety requirements of critical-use parts need to be addressed [10].

To satisfy this drive for FFF-processed parts in primary aerospace structures, stringent safety requirements must be implemented. The first is the need for fire-resistant materials that do not produce smoke when burned, produce fumes that are safe for users, and have a high melting temperature. The second requirement is to keep part porosity below 2% [11].

The PAEK family of thermoplastic materials (PAEK, PEKK, PEEK, …) complies with the specified safety regulations but is notoriously hard to use in Fused Filament Fabrication [12], requiring a heated chamber for the coalescence between filaments to occur. By increasing this temperature, the capillary time during which the coalescence takes place is longer, resulting in higher part strength [13].

During the deposition process, two adjacent beads of material are placed in intimate contact with each other. The high temperature of printing starts the bond formation process between filaments, which is the fusing of polymer chains at a molecular scale. When the cooling step of the process is finished, the healing of the part is characterized by the interdiffusion of polymer chains, from one bead to the next [14].

This adhesion mechanism is shown in Figure 1.

The finished parts have varying levels of coalescence, causing interrogations regarding the structural soundness of the part and hence safety requirements regarding part porosity.

This phenomenon was first modelled by Frenkel and Eshelby in 1945 and 1949 [16]. They described the coalescence of two spherical beads of material modelled as an incompressible and viscous Newtonian fluid in isothermal conditions with a uniform stress tensor. This problem was solved with a small angle approximation.

Pokluda et al., in 1997 [17], expended the Frenkel–Eshelby model for all angles with a Runge–Kuta numerical method to solve the differential heat equation. This model provides a dimensionless solution to the problem of neck growth of PLA beads. From this, the expected porosity should not vary with a change in the radius of the considered beads.

As shown by Bakrari Balani [18], the surface tension of the material changes during the cooling process and so does the viscosity. An accurate characterisation of both is needed to model the behaviour of the porosity of a printed part.

Lepoivre et al., in 2021 [19], characterized the evolution of the surface tension and viscosity of PEKK with regard to the temperature and then used a numerical model to describe the evolution of the degree of coalescence with varying surface tension and viscosity. Their findings are used to accurately calculate the surface tension and viscosity of the PEKK for the presented model.

In 2023, Jiang et al. [20] investigated the coalescence of PLA in FFF by modelling the behaviour of the coalescence and testing the results on printed filaments of varying sizes, but the long diameter of the filaments studied was limited to a nozzle diameter of 0.4 mm.

There is a need to determine the behaviour of larger filaments, as the increase in cooling time of the filaments increases with each increment in diameter. The effects of the chamber temperature, a parameter with a sure impact on a part’s Young modulus, are not often seen in the literature [21].

In this publication, a framework based on Jiang et al.’s investigations, with Lepoivre et al.’s data on the viscosity and surface tension of PEKK, explores the neck growth of PEKK filaments. This study will show the impact of nozzle diameter and cooling temperature on the longitudinal neck growth between filaments. The authors have not found other examples in the literature of similar studies for the considered nozzle diameter.

These investigations on the neck growth will also be conducted in different chamber temperatures to quantify the effect of the parameter on the porosity of the part.

## 2. Materials and Methods

This section pertains to the methodology used to analyse section cuts of printed PEKK test pieces.

### 2.1. Modelling of the Temperature of the Bead

The cooling of a single filament is simplified to a one-dimensional transient heat transfer model. This method works under the assumption of a uniform temperature distribution throughout the cross section of the deposited filament [16]. The nozzle moves at a constant speed *v*. The temperature of the substrate is considered constant throughout the deposition and cooling process, from which the conduction of heat between layers is considered convection. Figure 2 shows the energy interaction on a finite element dx of a heated filament.

From this, the heat differential equation of the system can be written.
(1)$$\rho CA\frac{\partial T}{\partial t}=A\frac{\partial (k\frac{\partial T}{\partial x})}{\partial x}-h\left(T-T_{\infty }\right)$$

Where $T=T_{0}$ at x=0, t≥0 and $T=T_{\infty }$ at x=∞ and t≥0, and *C* is the specific heat coefficient, ρ is the density, *A* is the cross section area of the filament, *k* is the heat conductivity coefficient and *h* is the convection coefficient of the system.

The coefficient *h* governs the effects of both heat convection with air and conduction with the substrate.

The nozzle is moving at a velocity *v* and x=vt. The time-dependent term $\frac{\partial T}{\partial t}$ can be rewritten as
(2)$$\frac{\partial T}{\partial t}=\frac{\partial T}{\partial x}\frac{\partial x}{\partial t}=\frac{\partial T}{\partial x}v$$

The previous equation then becomes:(3)$$\rho CAv\frac{\partial T}{\partial x}=A\frac{\partial (k\frac{\partial T}{\partial x})}{\partial x}-h\left(T-T_{\infty }\right)$$

And has the solution:(4)$$T\left(x\right)=T_{\infty }+\left(T_{0}-T_{\infty }\right)e^{-mx}$$

Where $m=\frac{\sqrt{1+4\alpha \beta }-1}{2\alpha }$, x=vt, $\alpha =\frac{k}{\rho Cv}$, and $\beta =\frac{hP}{\rho CvA}$. $P=\pi \left(a+b\right)\left(\frac{64-3\lambda^{4}}{64-16\lambda^{2}}\right)$ and A=πab, with a as the major axis and b the minor axis of the cross section of the deposited filament, considered elliptic, and where $\lambda =\frac{a-b}{a+b}$.

### 2.2. Modelling of Coalescence between Two Beads of Material

The porosity of the part is related to the sintering of two adjacent deposited filaments. The polymer filaments are printed at a temperature higher than the glass transition temperature. At those temperatures, the polymer is considered a Newtonian fluid.

Polychronopoulos and Vlachopoulos [23], in 2020, studied the role of heating and cooling in the sintering of both spheres and cylinders for additive manufacturing, detailing the steps to solve the coalescence problem of Fused Filament Fabrication.

The sintering process or coalescence is illustrated in Figure 3.

The sintering is the behaviour that two adjacent liquid beads follow to form one unique bead of a longer radius. It is a phenomenon that is dependent on the surface tension of the fluid, Γ, and the viscosity of the fluid η. The sintering process occurs until the temperature of the material crosses below the glass transition temperature as highlighted in the previous section. The material is then considered solid and will not be subject to further deformations related to the sintering of the beads. The sintering process is modelled from the dynamic equilibrium between the work of the surface tension forces and the work of the viscosity forces in the given fluid.

In the following sections, the equations leading to the expression of the evolution of the half angle of coalescence θ are reported. It is obtained in the form of a differential equation.

From θ, geometry relations between quantities give the theoretical longitudinal neck growth—the quantity of interest.

#### 2.2.1. Sintering of Spheres

If two spheres of radius $a_{0}$ intersect at one point of contact, they will gradually coalesce into one sphere of radius $a_{f}$ with $a_{f}=a_{0}\sqrt[{3}]{2}$, with that point of contact as the centre of the newly formed sphere.

As shown in Figure 3, at a given time *t*, the values of the radius of the bead a(t), the half angle of coalescence θ(t), and the longitudinal neck growth x(t) will be referred to as *a*, θ and *x*, respectively, in the following.

Pokluda et al. [17] established a framework to model the behaviour of the sintering of beads. The radius *a* of the beads is given by the conservation of the mass of the polymer, with the density of a bead constant with the temperature.
(5)$$a=a_{0}{\left(\frac{4}{{\left[1+\cos\theta \right]}^{2}\left[2-\cos\theta \right]}\right)}^{1/3}$$

The area of a section *S* of two beads at a given time is given by Equation (Equation 6).
(6)$$S=4\pi a^{2}\left[1+\cos\theta \right]$$

The flow of material for spherical bead sintering is given by a biaxial extensional flow field. This leads to solving the dynamic equilibrium between the work of viscous forces and surface tension, the expression of a differential equation θ, and the half angle of the coalescence of the beads.

Polychronopoulos et al. [23] show the steps to the solution, and the expressions of the quantities of interest θ, the half angle of coalescence, and y, the longitudinal neck growth between filaments, as follows. The differential equation in θ is solved using a Runge–Kuta algorithm with an initial value of θ, $\theta_{0}=0.01$ to avoid numerical instabilities.
(7)$$\dot{\theta }=\frac{\Gamma }{a_{0}\eta }\frac{2^{-5/3}\cos\theta \sin\theta {\left(2-\cos\theta \right)}^{1/3}}{\left(1-\cos\theta \right){\left(1+\cos\theta \right)}^{1/3}}$$

Finally, for spherical sintering, the longitudinal neck growth *y* is given by:(8)$$y=a_{0}\sin\theta {\left(\frac{4}{{\left(1+\cos\theta \right)}^{2}\left(2-\cos\theta \right)}\right)}^{1/3}$$

In the following sections, the half angle of coalescence and the longitudinal neck growth for spherical beads of material will, respectively, be noted $\theta_{sph}$ and $y_{sph}$

#### 2.2.2. Sintering of Cylinders

In the case of fused filament fabrication, the sintering process may also be modelled as two elliptic cylinders fusing, which can be characterized by two circular cylinders of a unique radius $r_{0}=b^{2}/a$ [20], with *a* the semi-major axis and *b* the semi-minor axis of the ellipse, as described in Figure 4. The ultimate result of such coalescence is a unique cylinder in which the temperature of the polymer is maintained high enough for the sintering to continue. The variation in the length of the cylinder will be ignored as it is negligible before the variation in the radius of the cylinder. For ease of comprehension and for the rest of the section, the functions r(t) (the radius of the two cylinders), θ(t) (the half angle of coalescence, as shown in Figure 4), and y(t) (the longitudinal neck growth of the process) will be referred to as *r*, θ, and *y*, respectively.

The area of a circular segment of a disk formed by the section of two cylinders is as follows [24] and is shown in Figure 5, where *R* is the radius, Θ is the central angle in radians equal to 2θ, c is the cord length, s is the arc length, h is the sagitta, d is the apothem, and $A_{seg}$ is the area of the segment (orange area in Figure 5):
(9)$$A_{seg}=\frac{R^{2}}{2}\left(\Theta -\sin\Theta \right)=R^{2}\left(\theta -\cos\theta \sin\theta \right)$$

From which the area of the section of two cylinders during the sintering process $A_{c}$ shown in orange in Figure 4 is given by: (10)$$A_{c}=2\left(A-A_{seg}\right)=2\pi r^{2}-2r^{2}\left(\theta -\cos\theta \sin\theta \right)=2r^{2}\left(\pi -\theta +\cos\theta \sin\theta \right)$$

From the conservation of mass between the steps of the sintering, the radius *r* of the circular cylinder is given [23]:(11)$$r=\frac{\sqrt{\pi }r_{0}}{\sqrt{\pi -\theta +\cos\theta \sin\theta }}$$

Where $r_{0}=\frac{b^{2}}{a}$ and θ is the half angle of the coalescence, as shown in Figure 4.

From the given expressions of nominal radius *r* and area $A_{c}$ and the dynamic equilibrium between the work of viscous forces and the surface tension, a differential equation with the half angle of coalescence θ is given.

The flow of material induced by the viscous forces and the surface tension of the fluid is a planar extensional flow when the sintering process is approaching the sintering of cylinders, meaning that the work of viscous forces is expressed as a function of the strain rate of the system. By definition, the strain rate and the velocity of the deformation are linked, and so the dynamic equilibrium between viscous forces and surface tension that exists in the system when the gravity is neglected results in a differential expression of θ, i.e., the half angle of coalescence.

Polycronopoulos et al. [23] describe the steps to the solution of this problem, and the quantities of interest θ and *y* are given by Equations (Equation 12) and (Equation 13). The differential equation in θ is solved using a Runge–Kuta algorithm with an initial value of θ, where $\theta_{0}=0.01$ to avoid numerical instabilities.
(12)$$\dot{\theta }=\frac{\Gamma }{2\sqrt{\pi }\eta r_{0}}\frac{\left[\left(\pi -\theta \right)\left(\cos\theta \right)+\sin\theta \right]{\left[\pi -\theta +\cos\theta \sin\theta \right]}^{\frac{1}{2}}}{{\left(\pi -\theta \right)}^{2}\sin\theta \tan\theta }$$
(13)$$y=r_{0}\frac{\sin\theta }{\sqrt{\pi -\theta +\cos\theta \sin\theta }}$$

In the following sections, θ and *y* for cylinder sintering will be referred to as $\theta_{cyl}$ and $y_{cyl}$.

### 2.3. Material

The material is the 60–40 amorphous ThermaX Polyether Ketone Ketone polymer (PEKK-A) formulated by 3DXTech in Grand Rapids, MI, USA and made using Arkema’s Kepstan [60/40 copolymer] fabricated in Mobile, AL, USA. It is amorphous and contains only PEKK. Following the manufacturer processing parameters, the printed temperature is 325–360 °C, and the chamber temperature is 70 to 150 °C. It has a glass transition temperature of 162 °C and a melting temperature of 335 °C.

### 2.4. Specimen Printing

The printed test specimens are created according to the 1BA Standard EN ISO 527-2 [25] dimensions. The .STL file of the corresponding geometry was sliced into a readable G-code for a funmat 610 HT printer fabricated by INTAMSYS in Shanghai, China.

The G-code path consists of superposed deposited filaments, as shown in Figure 6.

The values of the varying parameters of the factorial Design Of Experiment (DOE) used are given in Table 1. The DOE has three levels and two factors, where cooling temperature $T_{\inf}$ has the values 110, 130 and 150 °C, and the nozzle diameter *D* has the values 0.4, 0.6 and 0.8 mm. Printing failed for cooling temperatures below 110 °C. For each specified nozzle diameter *D* (mm) and chamber temperature $T_{\inf}$, a set of test pieces will be produced, resulting in nine parameter couples and samples. The layer thickness *H* of the part is given by H=D/2, as the optimum for print quality in the field is given for this value of *H* [26].

Other process parameters, such as printing temperature, also have an impact on the porosity of the part and are fixed to remain constant in the DOE. Test pieces were printed for each varying parameter couple. For *D*, the given nozzle diameter, and $T_{\inf}$, the given chamber temperature of a test piece, the denomination $X_{T_{\inf}}^{D}$ will be given to the test piece.

### 2.5. Specimen Preparation

The test specimens were then cut in the YZ axis using an Isomet 1000 precision cutter into 6 observable surfaces.

The cut pieces of the testing specimen were cured inside a transparent KM-U resin from Presi, made in Eybens, France.

The resin cylinders were then polished using a MECAPOL P200 polishing machine made by Presi in Eybens. The abrasive disc P180 was used to denude the surfaces of interest; then, the abrasive discs P360, P840, P1000, P2400 and P4000 were used successively for 5 min each, with water as a suspension. Subsequently, a TOP polishing disk was applied for 5 min with a 6 µm Reflex LDM suspension before a RAM polishing disk with a 3 µm Reflex LDM suspension.

The polishing machine head has a speed of 600 laps per minute with all disks and applies a pressure of 1.1 daN onto the polishing head. The resin cylinders were maintained as immobile during the process.

### 2.6. Mesostructure Observation

The samples were observed using a VHX-7100 microscope, made by Keyence in Osaka, Japan, where two captures of each surface of interest were taken. The lighting of the scene was made by coaxial lights and then optimized thanks to the microscope software. The images were then binarised using image processing tools.

There are 6 such mesotructures available for each pair of chamber temperature and nozzle diameter.

The mesostructures were then cut by layers, where the longitudinal length of the voids was measured and subtracted to the height of the layer, as shown in Figure 7, to obtain the longitudinal neck growth of this particular element in the layer. This operation is repeated for all voids in the layer.

In Figure 7, the longitudinal neck growth is obtained by subtracting the longitudinal void length (represented in green) from the layer height (represented in orange).

The results of all the measurements are then plotted in the next section.

## 3. Results

The results of the various investigations are shown and are divided into three parts. The first section presents the results of the theoretical model used to predict the length of the neck between filaments, the second displays the experimental data measured using the microscope, and the third compares the results of the two previous sections. All the results are then discussed in the following section.

### 3.1. Theoretical Longitudinal Neck Length

Using the introduced parameters, the theoretical thermal temperature of the filament is given in Figure 8:

From this, the coalescence time $t_{c}$ given to the polymer to form bonds in each case is obtained. It is shown as the time at which the temperature of the filament crosses the glass transition temperature in Figure 8. This coalescence time is then used as a limit in the time integration to obtain the theoretical half angle θ by solving the differential Equation (Equation 7) for spherical sintering and (Equation 12) cylinder sintering. By definition, an amorphous polymer under its glass transition temperature is considered solid, but the conformal changes in segments are considered infinitely slow at the glass transition temperature. By offering the possibility to deform until the glass transition temperature, the assumption is that the deformation modelled may be superior to the experimental deformation. In this study, the viscosity of the polymer changes with the temperature of the polymer. Bellow 200 °C, the model shows a negligible increase in longitudinal neck growth, which is in agreement with the expected behaviour of the polymer, with the majority of the deformation taking place in the first instants after the deposition.

From Equations (Equation 8) and (Equation 13), the theoretical neck growth is then obtained. All the theoretical results have been compiled in Table 2.

### 3.2. Experimental Results

The pictures of the cross section of test pieces for different nozzle diameters were taken via microscope. An example of such mesostructure is shown in Figure 9.

After image binarisation and cropping, Figure 10 shows the mesostructure of parts made with nozzle diameters of 0.4 mm, 0.6 mm and 0.8 mm at different chamber temperatures.

This process resulted in more than 60 measures of longitudinal neck growth per DOE set.

### 3.3. Experimental and Theoretical Comparison

From the mesostructure, the longitudinal neck growth values are extracted, as described previously, and compiled by layer and chamber temperature in Figure 11 for nozzle diameter D=0.4 mm and Figure 12 for nozzle diameter D=0.6 mm.

To help visualize the data, it was plotted in the form of a box plot, where the limits of the box represent the first and third quartiles of the data, and the whiskers represent the limits of the data. Outliers were plotted separately beyond the whiskers of the plot. The mean value of the data for each layer and DOE set was plotted as the black line in the box. The standard deviation of the data *s* can be approximated through the following equation: $s=\frac{3}{4}\left(Q_{3}-Q_{1}\right)$, with $Q_{1}$ the first quartile and $Q_{3}$ the third quartile.

From the shape of the porosity shown in the experimental results displayed for nozzle diameter D=0.8 mm, the sintering of the part is not the principal phenomenon behind the porosity of the part, so the comparison with the theoretical model is not conclusive.

In the abscissa of the graphs, L is the abbreviation of layer. It is followed by the number of the considered layer, starting with 1 for the bottom layer in contact with the raster. T is the abbreviation of temperature and is followed by the value of the chamber temperature corresponding to that particular print.

In the following section, the term “theoretical longitudinal neck growth” will refer to both the cylinder- and spherical-modelled neck growths, representing an interval of possible neck growth measurements. The spherical calculation serves as the lower limit of the interval, and the cylinder calculation serves as the upper limit.

As shown in the previous section, the longitudinal neck growth is not the driving force behind the porosity observed for D=0.8 mm, and the results of the experimental measurements are not the longitudinal neck growth of the filaments. The image analysis of the printed parts with a nozzle diameter of D=0.8 mm is not relevant to the comparison of the model with experimental results.

For D=0.4 mm, for the first few layers, the coalescence measured is less than the theoretical value: this is due to two phenomena. Firstly, the absence of contact between filaments is observed, as shown in Figure 10c, leading to measured longitudinal neck growth equal to zero, unlike the predictions of the model. This behaviour is observed for elements printed with all chamber temperatures, but mostly for $T_{inf}=110$ °C and $T_{inf}=150$ °C. Secondly, the values of the non-zero elements is also lower than expected, which is due to the proximity to the heating bed. The assumption that the conduction of heat through the material does not change its temperature is false: the substrate has a similar mass to the deposited filament, and hence, changes in the temperature in the substrate are induced by the presence of the filament. Therefore, the cooling of the material is faster than expected, leading to a lower experimental longitudinal neck growth measured than the theoretical value.

For the middle to top layers, the theoretical and experimental estimations of the neck growth results show an overlap as the majority of experimental measurements are contained between the two boundaries defined by the models, but a large number of experimental measurements show a quasi-total neck growth. This is due to the change in geometry observed in the mesostructure: the void shape going from diamonds to triangles. In the literature, occurrences of this are linked with over-extrusion, as shown in Ghorbani et al. [27].

For D=0.6 mm, there is contact between filaments in the first layers for all temperatures. As observed for D=0.4 mm, the neck growth for the first deposited layers is reportedly smaller than that of the middle or top of the printed test piece. The observed voids are almost all triangle-shaped. A complete absence of voids is also observed in several areas of the section. Large defects are also starting to appear in the top layer, leading to poor surface quality of the part for chamber temperature $T_{inf}=150$ °C. The filaments deposited in the last layers have a rounder appearance than those deposited in the first layer, while the extrusion speed of the material has not changed.

## 4. Discussion

### 4.1. Physics Analysis

This section pertains to the analysis of the physics governing the results previously displayed. Two phenomena are discussed: the first is the filament shape, and the second is the coalescence between filaments.

#### 4.1.1. Filament Shape

The model considers a perfect ellipse or sphere on which the equations of the system are solved. The experiments show that the shape of the filament is not so simple to describe. First, the nozzle is a main contributor to the shape of the deposited filament. Second, the asymmetry of the deposited filament is due to small variations in the angle of the nozzle during the printing process.

For a nozzle diameter D=0.4 mm, and while the material is printed, the nozzle constrains the expansion of the material on the Z-axis by its presence. The excess material is pushed to the sides of the nozzle. As observed in Figure 10, the centre of mass of the filament is higher than the middle of the filament on the Z-axis, and the skinny part of the filament is at the bottom. This particular deformation is a result of the path of the nozzle, restraining the degrees of freedom of the deposited material.For nozzle diameter D=0.6 mm, the shape of the filament shows a centre of mass lower than the middle of the filament on the Z-axis, and the skinny part is at the top. As stated in the previous section, the aspect of the filament is rounder for filaments at the top of the print for cooling temperature $T_{\inf}=150$ °C.For nozzle diameter D=0.8 mm, the filaments in the first layer show little to no voids, and the flow of material induced by the high printing temperature was enough to change the behaviour of the deposited filament. Higher up in the test piece, the aspect ratio of the observable filaments is near 1, meaning that the nozzle does not constrain the expansion of material on the Z-axis: the substrate of the considered layer is lower than expected. A direct consequence is that the geometry of the part shows defects, and the mesostructure is no longer easily observable.Small variations in the angle of the nozzle during the deposition process result in a favoured direction of flow of material. The resulting filaments show asymmetry, with the centre of mass of the filament skewing to either right or left. This small change in shape may result in changes in the mesostructure of the complete test piece. This phenomenon is observable for all nozzle diameters and cooling temperatures.

#### 4.1.2. Coalescence between Filaments

The coalescence between filaments varies a lot with the process parameters and, during the process, with the number of layers deposited. Here, the different types of coalescence observed are shown: absence of coalescence, coalescence resulting in diamond-shaped voids, coalescence resulting in triangular-shaped voids, and coalescence resulting in no voids.

For nozzle diameter D=0.4 mm and cooling temperature T=110 °C or T=150 °C, the first layers of deposited filaments display instances of the absence of coalescence. The asymmetry observed in the previous section, resulting in an absence of contact between filaments, stalls the coalescence between neighbouring filaments.For nozzle diameter D=0.4 mm, the voids can be in the shape of diamonds, formed by the coalescence of two elliptic filaments from the previous layer, and two from the current layer, meaning that the neck growth is as modelled in the theoretical part. It is, nevertheless, not the only type of void shape observed.For nozzle diameters D=0.4 mm and D=0.6 mm, the voids are often in the shape of triangles, meaning that the bottom left with the top right filament or top left with bottom right filament have started the sintering process, leading to two voids of smaller size and of triangular shape, or just one triangular-shaped void, as one of the two disappears during the sintering process. This is due, in part, to the asymmetry of the deposited filaments but also to the flow of material observed for D=0.6 mm filaments. This is seen in Figure 10d–f as the filament bottom part is sintering with its neighbours, but the top part is not, creating the aforementioned voids, when voids are created.For nozzle diameter D=0.8 mm, the analysis of the coalescence is not relevant, as the main contributor of voids is not the coalescence but the printing defects observed that resulted from the flow of material.

### 4.2. Theoretical and Experimental Results Discussion

The theoretical models describe two flow states between spherical, corresponding to a local sintering of the filament with its neighbour, and cylinder, corresponding to the global sintering of the whole filament length with its neighbour. The size of the zone heated by the deposition of a filament in its neighbour is the parameter that governs the flow of material from one filament to the next. If the heated zone is small before the length of the filament, then the flow state will approach a spherical sintering (the contact with the neighbouring filament changes the kinematics of the problem, and the flow state is not strictly spherical). If the heated zone is comparable to the length of the filament, then the flow state will correspond to a cylinder-shaped sintering. The reality is supposedly a combination of the two.

The given models may produce limits for the porosity inside a printed part, as the results for D=0.4 mm and D=0.6 mm show a significant overlap between the theoretical and experimental results, with both the spherical and cylinder sintering predictions acting as limits to the experimental results. There are cases where the prediction is not as precise:In the first layers of the print, this behaviour is changed as the conduction of the heat between the filament and the substrate is faster than expected, leading to smaller longitudinal neck growths measured.For D=0.6 mm, the Biot number of the filament $Bi=\frac{h}{k}\frac{V}{A}$, is very similar to 0.1. The approximation giving the uniformity of the temperature in the section of the considered filament may not be verified and the actual temperature of the point of contact between filaments may be lower than modelled, leading to the overestimation of the longitudinal neck growth by the considered models.

The models provided are also constant throughout the printing process and do not take into account the layer number of the print, while the experimental results show a clear influence of the layer number and especially the first two deposited layers. There are also deposited filaments that have a very different aspect ratio than other filaments in the part. They are found in the higher layers of the print.

The theoretical longitudinal neck growth may be equal or greater than the layer height H=0.3 mm. While the experimental neck growth is bounded by the process parameters, as the method used to measure the experimental neck growth requires a void length to be subtracted from the layer height, it is not the case for the theoretical longitudinal neck growth. The cases in which the coalescence is such that there is no void between filaments were simply not measured. The rest are represented in Figure 12.

#### Impact of the Cooling Temperature on Porosity

While the models show a clear dependence on the cooling temperature of the process, the experimental results show similar levels of mean longitudinal neck growth for all cooling temperatures. The impact of the cooling temperature on the part is mainly seen in the variations in neck growth measured between layers.

The porosity of the printed parts shows little to no variation with regards to the cooling temperature, but the models do. While it is often true that the lower the cooling temperature, the smaller the neck growth, the variation in neck growth is small. Still, the spherical model is better at approximating the experimental data than the cylinder model, especially for $T_{\inf}=150$ °C, where both models predict higher values than the experimental data.

### 4.3. Hypothesis for the Observed Differences

In this section, the differences in results are explored. First, the differences between experimental results for different parameters are hypothesised. Secondly, the differences between the theoretical and experimental results are explained.

#### 4.3.1. Differences between Experimental Results for the Given Parameters

The differences described in the previous sections can be explained thanks to a few hypotheses. The differences mentioned can be grouped by the phenomena responsible for their existence. The first group of differences is as follows:The changes in aspect ratio observed for D=0.6 mm and D=0.8 mm but not for D=0.4 mm. The aspect ratio for the last layer for D=0.6 mm and the last few layers for D=0.8 mm is much closer to 1 than expected.When the observed aspect ratio of the filament, given by the ratio of $\frac{H}{D}$, is equal to 0.5, the position of the centre of mass of the filament changes with the nozzle diameter: the higher the nozzle diameter, the lower the centre of mass.

An explanation for these few differences might be the effect of gravity on the filament and the flow of material, causing the filament to “sag”. The changes in the position of the centre of mass of the filament show that a force is applied to the filament and acts on the direction of the flow of material. This force is negligible for D=0.4 mm but starts to affect the filament shape for D=0.6 mm.

A slight variation in layer height induced by gravity that is undetected by the nozzle and compounds at each layer, may also explain the variation in aspect ratio for such filaments. With the substrate at a lower position than expected by the nozzle, the plastic deformation induced by the nozzle will be lower than for the previous layers, leading to higher chances of defects and changes in the aspect ratio of the filament.

Another set of differences can be explained thanks to the preferred direction of the flow of material from the nozzle, induced by a slight angling defect of the nozzle:The coalescence of filaments from different layers that are not directly above one another that occurs systematically for D≥0.6 mm and often for D=0.4 mm.The lack of coalescence between filaments observed in the first few layers for D=0.4 mm.

As the nozzle is not completely parallel to the build plate, a small angle defect will impact the direction of the flow of filaments, skewing the position of filament centre toward the sides and creating an absence of coalescence in-between neighbouring filaments. In the case for coalescence between filaments that should not be in contact, the changes in geometry induced by this slight defect create now-neighbouring filaments to coalesce with, impacting the voids’ shape and length.

The last set of differences seen pertains to:The value of the longitudinal neck growth for non-zero elements of the first few layers being lower than expected.The changes observed for the longitudinal neck growth for different layers of the printed test piece.

This last set of differences is explained thanks to the changes in conduction that occur during the build of the part. In the first layers, the heated zone of the substrate by the filament may be longer than the substrate itself. However, the printing bed cools the material to a given value, making the filament cool faster than expected, as the heat given to the substrate is absorbed by the printing bed. After the first few layers, a plateau is reached and the longitudinal neck growth of the following layers shows less variation once the zone heated by the filament is farther away from the printing bed.

#### 4.3.2. Limits of the Models

The models, either describing the sintering of spheres or cylinders, are able to give an approximation of the values of the measured longitudinal neck growth of a print when compared with compatible data, but they are not able to account for all experimental variations.

This approximation is sometimes too high. This can be explained by the combination of two factors:For D=0.6 mm, the variation in temperature resulting in the variation in the model prediction to values around and above the layer height of the print H=0.3 mm. It is not represented in the experimental data because of the absence of voids for a large part of the observed section, resulting in no longitudinal neck growth to compare the model prediction to.The temperature inside a filament is not entirely constant, so the temperature at the points for contact may be lower than modelled, leading to smaller longitudinal neck growth measured.

Still, the majority of measurements of the neck growth are found between the upper limit given by the modelling of the sintering of cylinders and the lower limit given by the modelling of the sintering of spheres.

The variations in the longitudinal neck growth between layers are also not shown by the models, as the variations induced by the proximity of the printing bed and the detected aspect ratio variations are not studied. In general, unexpected experimental variations are not predicted by the model.

The variations in theoretical longitudinal neck growth recorded when varying the cooling temperature of the print were not measured experimentally. The experimental longitudinal neck growth shows little variation in absolute value with the cooling temperature, but there is variation in the layer number where the maximum longitudinal neck growth is recorded.

During the deposition, two phenomena occur at the same time:The heating of the neighbouring filament by the deposited filament.The cooling of both filaments by conduction and convection.

These two phenomena delimit a heated zone in which the sintering between filaments can occur. The shape of the heated zone as a result of the deposition of a filament is a strong indicator of which spherical or cylindrical model to use. If the cooling of the part is the dominating phenomenon, the shape of the heated zone will tend towards the sphere. Otherwise, the shape of the heated zone will tend towards the cylinder. Here, the spherical model shows a better approximation of the longitudinal neck growth than the cylindrical model, and the shape of the heated zone is closer to the sphere than to the cylinder.

While the used models provide an interesting approximation towards the solution, here are a few phenomena not considered that may have an impact on the longitudinal neck growth:The gradient of the temperature inside the filament and the dynamic changes in temperature resulting from such considerations.The effect of the previous layers on the actual layer.The consideration of voids of various shapes inside the mesostructure.

## 5. Conclusions

A numerical model of the coalescence between filaments was applied to PEKK for larger filament diameters than available in the literature. The impact of the nozzle diameter and chamber temperature on the longitudinal neck growth was measured using image analysis. The mesostructure variations observed were discussed and the differences in the numerical models explained, furthering the knowledge of the behaviour of high-performance polymer parts made by Fused Filament Fabrication. The following conclusions are given:While the order of magnitude of the model is correct, the experimental nature of fused filament fabrication leads to a high disparity in neck growths inside a single layer, leading also to a disparity in void size between layers.When the shape of the void is in accordance with the model, the model can be used to obtain a quantitative approximation of the longitudinal neck growth when knowing the printing parameters. The instances where the shape of the void is not in accordance with the model are when the absence of contact between filaments is observed or when the voids are not in the shape of diamonds, as shown in Figure 10.In a perfect environment, the impact of the nozzle diameter will easily be explained thanks to Equations (Equation 7) and (Equation 12). The plastic deformation of the filament by the nozzle changes the geometry of the filament, leading to disparities between the theoretical model and the experimental measures.The spherical model acts as the upper boundary of the porosity inside a part. As the porosity of a part is limited, a conservative approach to the calculation of the neck growth results in the higher porosity of a part, leading to stricter security margins, as discussed previously in the comparison between experimental and theoretical neck growth and shown in Figure 11 and Figure 12.

To further the prediction of the effect of chamber temperature and nozzle diameter on the longitudinal neck growth, a complete study on computational fluid dynamics should be carried out with the goal of solving the Navier–Stokes equation of the coalescence of two adjacent filaments to compare to the experimental data and validating the approximations used in the model.

A model of the neck growth of triangle-shaped voids should be added to the model to predict the neck growth in the test piece when relevant. A diagnostic of when to use diamond- or triangle-shaped voids should be added to the model.

To link the porosity of the part to its mechanical behaviour, mechanical testing was conducted on the printed parts, and further work to link printing parameters and mechanical behaviour is in progress. Then, a framework for printing parameters, porosity, and mechanical behaviour will be available for use.

The road to the zero-waste fabrication objective of additive manufacturing for high-end polymers starts with our understanding of the porosity of the part. Computational fluid dynamics models, coupled with high-resolution cameras, automatic defect detection algorithms and machine learning may result in instant parameter optimisation during the fabrication process, leading to fewer defects and less waste.

## Figures and Tables

**Figure 1 polymers-16-02732-f001:**
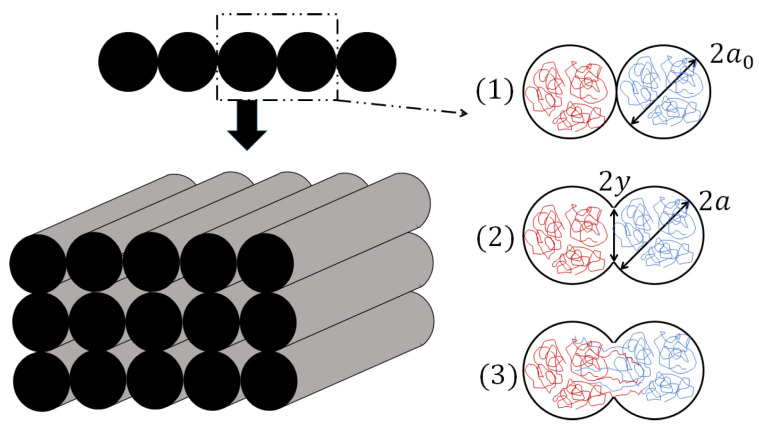
Bond formation process between two filaments: (1) surface contact; (2) neck growth; (3) molecular diffusion at interface and randomisation—inspired from [15].

**Figure 2 polymers-16-02732-f002:**
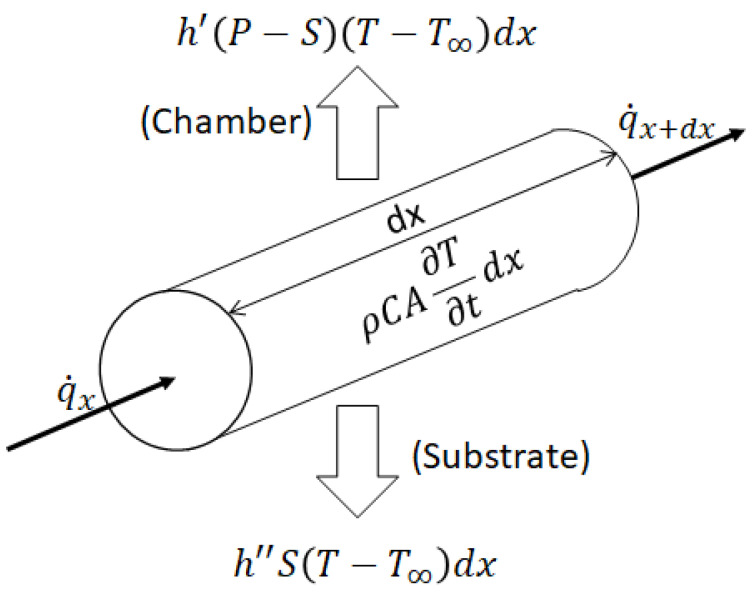
Energy interaction on finite element dx, inspired by [22].

**Figure 3 polymers-16-02732-f003:**
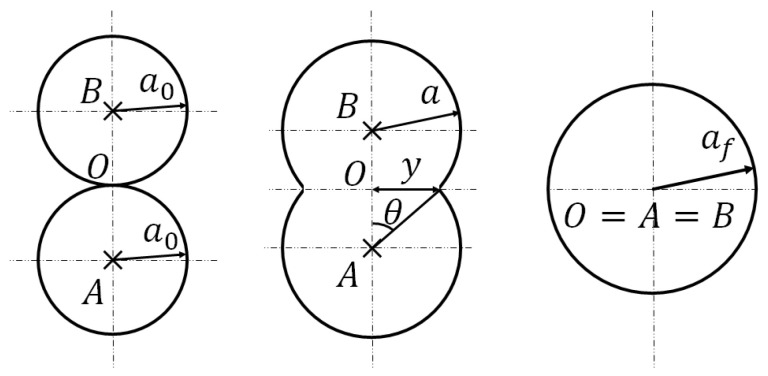
Sintering process of two cylinders or two spheres, inspired by [23].

**Figure 4 polymers-16-02732-f004:**
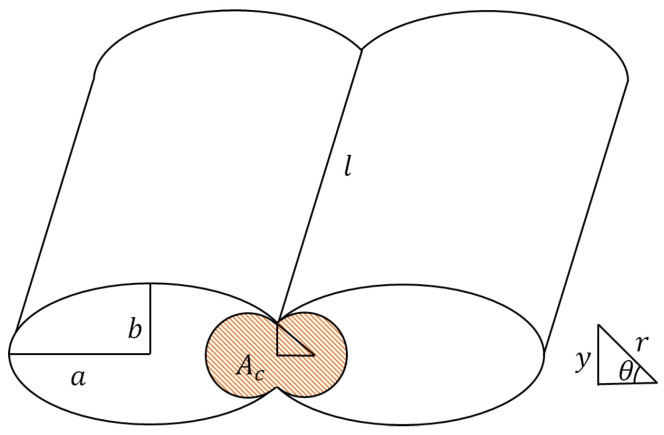
Diagram of bonding, inspired by [20].

**Figure 5 polymers-16-02732-f005:**
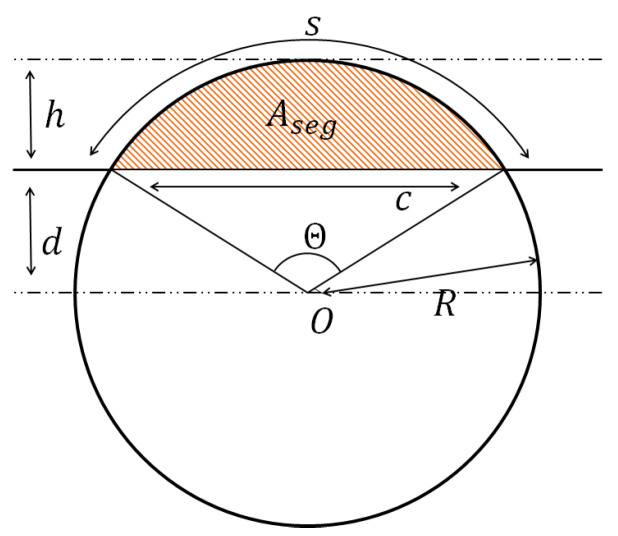
Area of a segment of a circle.

**Figure 6 polymers-16-02732-f006:**
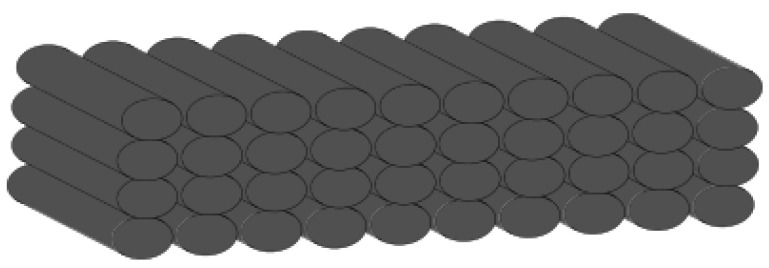
G-code path.

**Figure 7 polymers-16-02732-f007:**
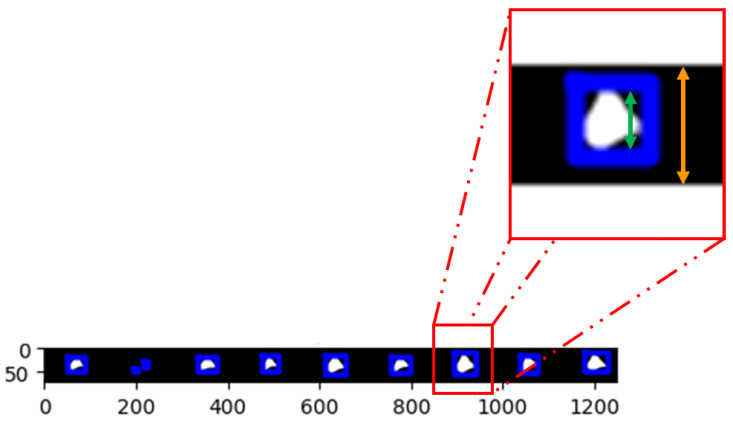
Measurement of longitudinal neck growth of one coalescence occurrence in a designated layer. The colour lengths are the layer height (orange) and the longitudinal void length (green).

**Figure 8 polymers-16-02732-f008:**
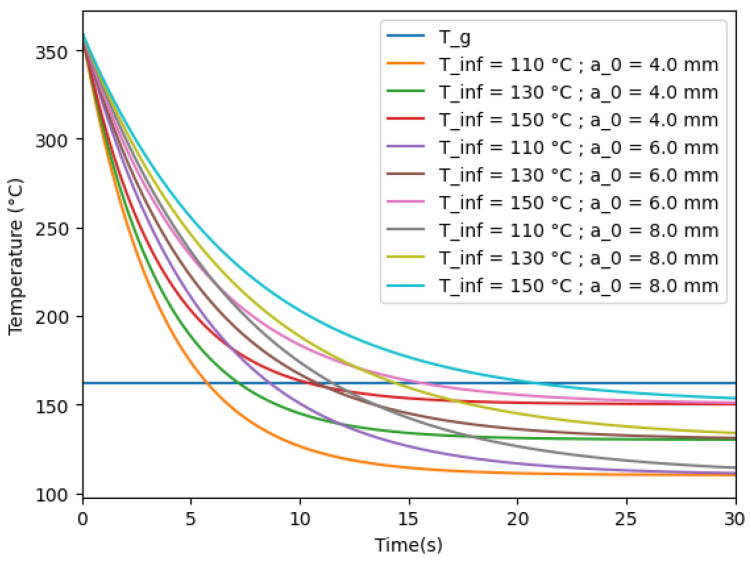
Theoretical temperature of the deposited filament.

**Figure 9 polymers-16-02732-f009:**
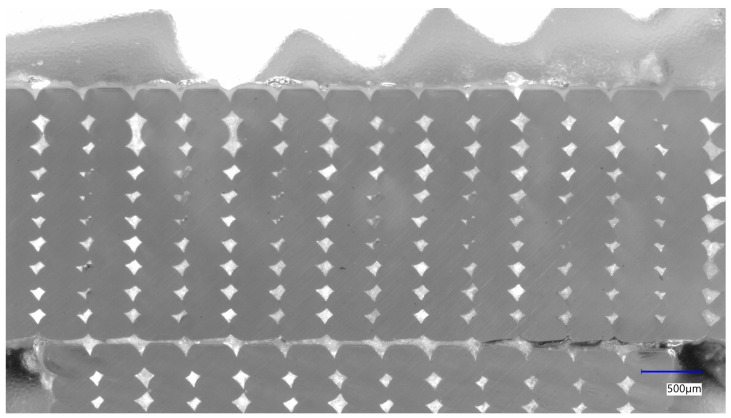
Microscope image for 0.4 mm nozzle, 110 °C chamber temperature: $X_{110}^{0.4}$.

**Figure 10 polymers-16-02732-f010:**
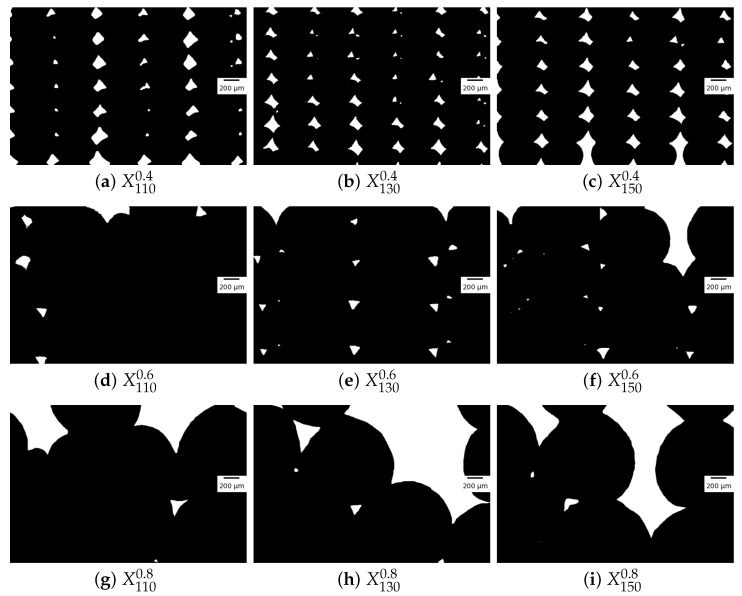
Examples of mesostructures of printed test pieces at different chamber temperatures $T_{\inf}$ and nozzle diameters *D* noted $X_{T_{\inf}}^{D}$.

**Figure 11 polymers-16-02732-f011:**
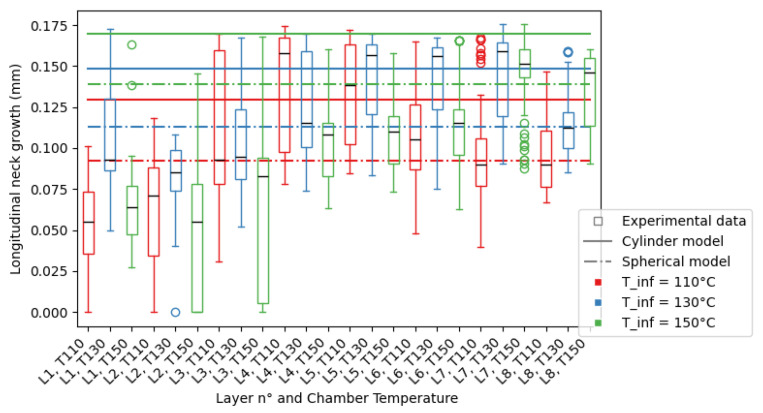
Neck growth for nozzle diameter D=0.4 mm, for all layers and cooling temperatures ($T_{\inf}=110$ °C in red, $T_{\inf}=130$ °C in blue, $T_{\inf}=150$ °C in green), for experimental data (boxplot), and models (spherical model as dotted line and cylinder model as line).

**Figure 12 polymers-16-02732-f012:**
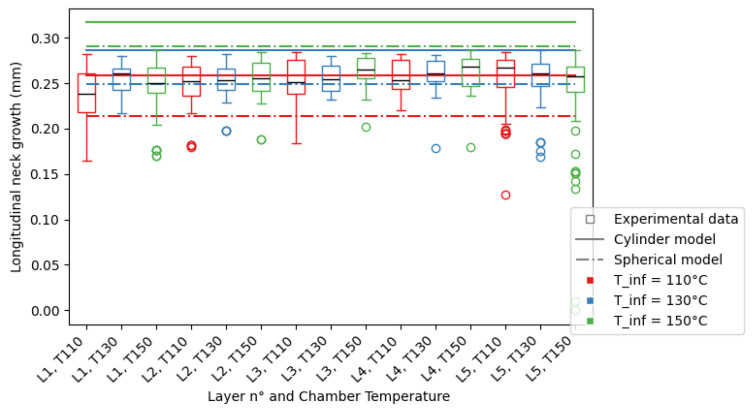
Neck growth for nozzle diameter D=0.6 mm, for all layers and cooling temperatures ($T_{\inf}=110$ °C in red, $T_{\inf}=130$ °C in blue, $T_{\inf}=150$ °C in green), for experimental data (boxplot), and models (spherical model as dotted line and cylinder model as line).

**Table 1 polymers-16-02732-t001:** Test piece DOE.

Test Piece	$X_{110}^{0.4}$	$X_{130}^{0.4}$	$X_{150}^{0.4}$	$X_{110}^{0.6}$	$X_{130}^{0.6}$	$X_{150}^{0.6}$	$X_{110}^{0.8}$	$X_{130}^{0.8}$	$X_{150}^{0.8}$
$T_{\inf}$ (°C)	110	130	150	110	130	150	110	130	150
*D* (mm)	0.4	0.4	0.4	0.6	0.6	0.6	0.8	0.8	0.8
*H* (mm)	0.2	0.2	0.2	0.3	0.3	0.3	0.4	0.4	0.4

**Table 2 polymers-16-02732-t002:** Theoretical results of capillary time $t_{c}$, theoretical half angle of coalescence $\theta_{cyl}$ and $\theta_{sph}$, theoretical longitudinal neck growth $y_{\mathrm{cyl}}$ and $y_{\mathrm{sph}}$ for both spherical and cylinder sintering.

Test Piece	X1100.4	X1300.4	X1500.4	X1100.6	X1300.6	X1500.6	X1100.8	X1300.8	X1500.8
Tinf (°C)	110	130	150	110	130	150	110	130	150
*D* (mm)	0.4	0.4	0.4	0.6	0.6	0.6	0.8	0.8	0.8
*H* (mm)	0.2	0.2	0.2	0.3	0.3	0.3	0.4	0.4	0.4
$t_{c}$ (s)	5.82	7.22	10.43	8.73	10.83	15.64	11.53	14.44	20.85
$\theta_{\mathrm{cyl}}$ (rad)	0.26	0.3	0.35	0.35	0.39	0.43	0.41	0.45	0.49
$\theta_{\mathrm{sph}}$ (rad)	0.18	0.22	0.27	0.28	0.33	0.38	0.36	0.4	0.45
$y_{\mathrm{cyl}}$ (mm)	0.13	0.15	0.17	0.26	0.29	0.32	0.41	0.44	0.48
$y_{\mathrm{sph}}$ (mm)	0.09	0.11	0.14	0.21	0.25	0.29	0.36	0.41	0.47

## Data Availability

Data available upon request.

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
