# Peer review of "Investigation on the Coalescence of Filaments in Fused Filament Fabrication Process for Amorphous Polyether Ketone Ketone for Varying Nozzle Diameters and Chamber Temperatures"

_polymers, 2024, doi:10.3390/polym16192732_

Round 1
Reviewer 1 Report
Comments and Suggestions for Authors
This study examines the impact of printing parameters, specifically nozzle diameter and cooling temperature, on the porosity of parts produced through fused filament fabrication of polyether ketone ketone polymer. Utilizing existing modeling methods based on the sintering of cylinders and spheres, the authors compare model predictions with microscopic images of part sections to assess porosity. The findings indicate that while cooling temperature minimally affects porosity, both nozzle diameter and layer thickness significantly influence mesostructural characteristics.
This is an intriguing study that addresses an important topic; however, several key issues must be resolved before a final decision can be made.
1. Including the abbreviations in the title is not recommended. Please provide the full forms of FFF (fused filament fabrication) and PEKK-A (amorphous Polyether Ketone Ketone polymer) in the title. The title can be modified as “Investigation on the coalescence of filaments in fused filament fabrication process for polyether ketone ketone polymer at varying nozzle diameters and chamber temperatures”.
2. Given the discrepancies observed between the theoretical and experimental longitudinal neck growth measurements, particularly for the nozzle diameters of D = 0.4 mm and D = 0.6 mm, how do you plan to refine your model to account for the effects of filament contact, heat conduction from the substrate, and the geometric changes in void shapes, and what implications might these refinements have on the overall understanding of porosity in fused filament fabrication?
3. Please elaborate on the assumptions made in your theoretical model regarding the relationship between coalescence time tc​ and the glass transition temperature, and how these assumptions may affect the accuracy of the theoretical half angle θ and neck growth predictions for both spherical and cylindrical sintering?
4. Considering the observed discrepancy between the models and experimental results regarding the influence of cooling temperature on mean longitudinal neck growth and porosity, what specific factors do you believe contribute to the spherical model's superior approximation of the experimental data compared to the cylindrical model, and how might these insights inform future modeling approaches in fused filament fabrication?
5. The abstract requires significant revision. It should include the following components: i) the significance of the topic and/or a reference to existing literature and/or the identification of a knowledge gap; ii) the objectives of the current study; iii) a description of the methods employed; iv) a summary of the key findings; and v) the implications of these findings and/or the value of the study. However, the current version of the abstract lacks both the fourth and fifth components.
6. Figure 1 does not provide specific information. Please eliminate it.
Reviewer 2 Report
Comments and Suggestions for Authors
Very interesting research and modelling. But I have few remarks/questions/suggestions:
a) Is there any reason for using test specimen for ISO 527-2 - tensile properties while you are not making any research in this field (tensile properties)?
b) What type of DOE you applied in your research? Factorial?
c) I can not see any repetition of your DOE sets which can give you an information on method error/noise? Generally in the middle of the design.
d) Did you produce more specimens with each of DOE sets? If yes, please provide information about standard deviations.
e) You connected layer thickness directly with nozzle diameter. But you can also produce different layer thicknesses with the same nozzle. In many researches layer thickness is showed as a significant parameter of FFF - maybe you should consider it as a third independent parameter in your future research. In this paper it is not clear what is the contribution of the layer thickness to the obtained results.
f) You select the lowest chamber temperature od 110 degrees although the recommended temperature from the material producer is 70 degrees. Is there any reason for this?
g) Do you think that nozzle temperature does not have significant influence on the coalescence?
Round 2
Reviewer 1 Report
Comments and Suggestions for Authors
This revised paper can be accepted. The authors have addressed all the comments and enhanced paper.